# PKCθ links proximal T cell and Notch signaling through localized regulation of the actin cytoskeleton

Graham J Britton[1], Rachel Ambler[1], Danielle J Clark[1], Elaine V Hill[1], Helen M Tunbridge[1], Kerrie E McNally[1], Bronwen R Burton[1], Philomena Butterweck[1], Catherine Sabatos-Peyton[1], Lea A Hampton-O'Neil[1], Paul Verkade[2], Christoph Wülfing[1*†], David Cameron Wraith[1*†‡]

[1]School of Cellular and Molecular Medicine, University of Bristol, Bristol, United Kingdom; [2]School of Biochemistry, University of Bristol, Bristol, United Kingdom

**\*For correspondence:** Christoph. Wuelfing@bristol.ac.uk (CW); d.wraith@bham.ac.uk (DCW)

†These authors contributed equally to this work

**Present address:** ‡Institute of Immunology and Immunotherapy, University of Birmingham, Birmingham, United Kingdom

**Competing interests:** The authors declare that no competing interests exist.

**Abstract** Notch is a critical regulator of T cell differentiation and is activated through proteolytic cleavage in response to ligand engagement. Using murine myelin-reactive CD4 T cells, we demonstrate that proximal T cell signaling modulates Notch activation by a spatiotemporally constrained mechanism. The protein kinase PKCθ is a critical mediator of signaling by the T cell antigen receptor and the principal costimulatory receptor CD28. PKCθ selectively inactivates the negative regulator of F-actin generation, Coronin 1A, at the center of the T cell interface with the antigen presenting cell (APC). This allows for effective generation of the large actin-based lamellum required for recruitment of the Notch-processing membrane metalloproteinase ADAM10. Such enhancement of Notch activation is critical for efficient T cell proliferation and Th17 differentiation. We reveal a novel mechanism that, through modulation of the cytoskeleton, controls Notch activation at the T cell:APC interface thereby linking T cell receptor and Notch signaling pathways.

## Introduction

T cell activation is mediated by antigen recognition through the T cell receptor (TCR). To allow physiological adaptation, the TCR signal is modulated by co-regulatory signals. Here, we address co-stimulation through Notch. Notch family proteins are large, heterodimeric transmembrane receptors. Upon Notch ligation by one of a family of Notch ligands (*Osborne and Minter, 2007*), a plasma membrane-tethered matrix metalloproteinase, ADAM10 or ADAM17, removes the Notch extracellular domain. Subsequently, the plasma membrane-embedded γ-secretase complex liberates the Notch intracellular domain (NICD). The NICD constitutively translocates to the nucleus where it displaces transcriptional repressors and recruits enhancers to genomic loci characterized by binding of the transcription factor RPBJκ (*Borggrefe and Oswald, 2009*).

An essential role for Notch1 in T cell thymic development is well established (*Robey et al., 1996*; *Washburn et al., 1997*); a Notch1 deficient hematopoietic compartment yields no T cells (*Radtke et al., 1999*). Notch1 signaling also plays a pivotal role in mature T cells. Notch1 is activated following TCR stimulation (*Amsen et al., 2004*; *Guy et al., 2013*; *Ong et al., 2008*) and is required for effector cell development (*Amsen et al., 2004*; *Keerthivasan et al., 2011*). The degree of Notch1 activation is directly proportional to the strength of the TCR signal (*Guy et al., 2013*). Antigen-induced Notch1 activation in T cells may be ligand independent (*Adler et al., 2003*) (*Ayaz and Osborne, 2014*; *Palaga et al., 2003*). However, the cellular mechanism coupling proximal T cell signaling to Notch activation is unresolved. Here we reveal a spatially constrained mechanism of Notch1 activation. We demonstrate that PKCθ, a serine/threonine kinase integrating TCR and

**eLife digest** The body's immune system recognizes and responds to foreign agents such as bacteria and viruses. Immune cells known as T cells recognize foreign substances through a protein on their surface called the T cell receptor. Specifically, the T cell receptor binds to fragments of foreign proteins displayed on the surface of other cells, which sets in motion a chain of events that leads to the T cell becoming activated. An activated T cell divides to form new cells that develop into "effector" T cells, which can mount an effective immune response.

The T cell engages with the cell displaying the foreign proteins via an interface referred to as the immunological synapse. This zone of contact brings together the signaling machinery of the T cell. Like many other cells, T cells contain an internal skeleton-like structure made up of actin filaments. These filaments are crucial for the formation of the immunological synapse, in part because they help to transport the T cell receptor and other signaling proteins to the immunological synapse.

Recent research suggests that a signaling protein called Notch plays an important role in instructing activated T cells to develop into effector cells. Notch is found on the surface of many cells, including T cells, and it becomes activated when it is cut by a specific enzyme. However, it was not entirely clear how T cell signaling drives the activation of the Notch protein.

Britton et al. have now investigated the mechanism that leads to Notch activation in T cells from mice. The results show that a protein found inside the T cell, called PKCθ, is a major contributor to Notch activation when T cells become activated. So how does the PKCθ protein control the activation of Notch? Britton et al. observed that PKCθ inactivates a protein that normally inhibits actin filaments from forming, and does so specifically at the center of the immunological synapse. This inhibition promotes the generation of a large actin-rich structure known as the lamellal actin network. This structure is required to recruit the Notch-cutting enzyme to the immunological synapse. Further analysis revealed that Notch gets cut and activated during the first few minutes of T cell activation leading to cell division and the development of effector T cells.

Following on from this work, the next challenge will be to explore if altering signaling from the T cell receptor – for example, using drugs or small molecules – can modify the activation of Notch. If so, it will be important to explore if the chemicals could potentially be used to treat diseases that develop when T cells go awry, such as rheumatoid arthritis, psoriasis and Crohn's disease.

CD28 signals (*Altman and Kong, 2016*), enhances T cell actin dynamics through localization and phosphorylation of the negative actin regulator Coronin1A (Coro 1A) and thus mediates actin-based recruitment of ADAM10 to the T cell:APC interface for efficient Notch activation.

## Results and discussion

### PKC$\theta$ enhances Notch activation

To study the role of TCR/CD28-proximal signaling in Notch1 activation, we bred PKCθ-deficient Tg4 mice (Tg4$^{KO}$). PKCθ integrates TCR and CD28 signals. PKCθ-deletion renders peripheral T cells hyporesponsive but allows normal thymic selection (*Sun et al., 2000*). Tg4 CD4$^+$ T cells (*Liu et al., 1995*) recognize the acetylated N-terminal peptide of myelin basic protein Ac1-9[4K] and its high affinity MHC-binding analogue Ac1-9[4Y].

To determine whether Notch activation could play a role in mature T cells that is comparable to that in thymocytes, where Notch drives critical developmental decisions (*Radtke et al., 2013*), we assayed NICD expression in Tg4 thymocytes, naïve and primed T cells in response to anti-CD3 and anti-CD28. NICD expression and changes thereof upon cellular activation were similar (*Figure 1—figure supplement 1A and B*). However, Notch activation was impaired in mature T cells from Tg4$^{KO}$ mice. Tg4$^{KO}$ mice showed reduced Notch1 expression sixteen hours after in vivo T cell activation by injecting mice with 80 µg [4Y] peptide s.c. (*Figure 1A,B*) even though Tg4$^{KO}$ mice were grossly normal with a reduced number and proportion of CD4$^+$ splenocytes but unaffected Tg4 TCR expression (*Figure 2A*; *Figure 2—figure supplement 1*). Reduced Notch expression was confirmed by Western blot analysis 60 min after s.c. administration of [4Y] peptide (*Figure 1C*) and through

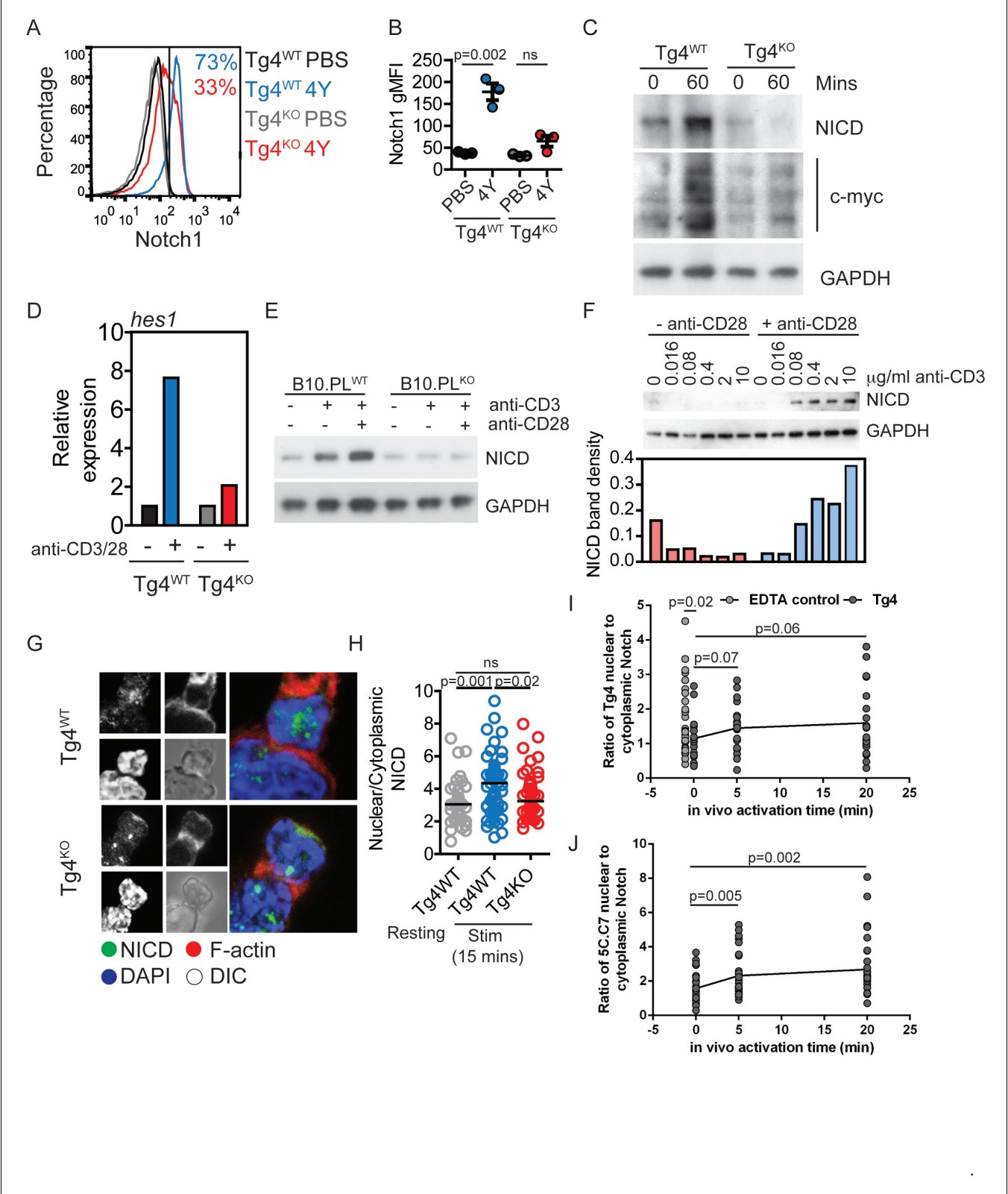

**Figure 1.** PKCθ enhances antigen-induced Notch activation. (**A**) Tg4[WT] and Tg4[KO] mice were injected subcutaneously with 80 μg of MBPAc1-9[4Y] peptide or PBS. After 18 hr splenocytes were immunostained to assess intracellular Notch1 expression and analyzed by flow cytometry. Gated on live, CD4[+] cells. (**B**) The expression (geometric mean fluorescence intensity, gMFI) of intracellular Notch1 in CD4[+] T cells from spleens of Tg4[WT] and Tg4[KO] mice treated as in **A** is shown as the mean ± SEM. **p=0.002, ns p=0.06 (ANOVA). One experiment of 2, n = 3 mice per condition. (**C**) Tg4[WT] and

*Figure 1 continued on next page*

*Figure 1 continued*

Tg4^{KO} mice were injected subcutaneously with 80 µg of MBPAc1-9[4Y] peptide or PBS. CD4^{+} T cells were isolated from the spleen after 60 min by MACS and protein extracts analyzed by Western blotting with anti-NICD, anti c-myc and GAPDH. One representative Western blot of three. (**D**) Naïve Tg4^{WT} and Tg4^{KO} CD4^{+} T cells were isolated from splenocytes and stimulated with plate-bound anti-CD3 and anti-CD28 for 30 min. Expression of Hes1 was determined by RT-PCR. One representative experiment of four. (**E**) Naïve CD4^{+} T cells were isolated from B10.PL PKCθ WT or KO splenocytes by magnetic selection and stimulated for 18 hr with plate-bound anti-CD3 and anti-CD28 as indicated. An equal amount of protein extract from each sample was analyzed for expression of the NICD and GAPDH by Western blotting. (**F**) Naïve CD4^{+} T cells isolated from Tg4^{WT} and Tg4^{KO} mice were stimulated for 18 hr with a titration of plate-bound anti-CD3 ±2 µg/ml anti-CD28, as indicated. Expression of NICD and GAPDH was assessed by Western blotting. One representative Western blot of two. (**G, H**) Tg4^{WT} and Tg4^{KO} T cell blasts were incubated for 15 min with [4Y]-loaded PL8 cells before fixation and immunostaining against the IC domain of Notch1. The cells were counterstained with phalloidin and DAPI before imaging by confocal microscopy. The proportion of NICD staining in the nucleus (defined by DAPI staining) and the cytoplasm (defined by phalloidin staining) was measured and the ratio of nuclear:cytoplasmic NICD calculated. **$p=0.0014$, *$p=0.02$, ns $p=0.2$ (ANOVA). 32–58 cells analyzed per condition, combined data from two independent experiments. (**I**) Tg4^{WT} mice were injected subcutaneously with 80 µg of MBPAc1-9[4Y] peptide or PBS. CD4^{+} T cells were isolated from the spleen after 5 or 20 min by MACS, fixed and immunostained against the IC domain of Notch1. The ratio of nuclear:cytoplasmic NICD is given. T cell treatment with 2 mM EDTA serves as a positive control of Notch activation. The difference between the 0 min time point and the EDTA control is significant with $p=0.02$ (ANOVA). One representative experiment of 4. (**J**) 5C.C7 mice were injected subcutaneously with 80 µg of MCC (89–103) peptide or PBS. CD4^{+} T cells were isolated from the spleen after 5 or 20 min by MACS, fixed and immunostained against the IC domain of Notch1. The ratio of nuclear:cytoplasmic NICD is given. Differences between the 0 versus 5 and 20 min time points are significant with $p=0.005/0.002$, respectively (ANOVA). One representative experiment of 3.

The following figure supplement is available for figure 1:

**Figure supplement 1.** NICD expression is comparable across Tg4 thymocytes, naïve and primed T cells and substantially enhanced upon retroviral expression.

analysis of Notch1-dependent hes1 expression (*Figure 1D*). Corroborating these data in non-TCR transgenic T cells, Notch1 cleavage in naïve CD4^{+} T cells (CD4^{+}, CD44^{−}, CD25^{−}) from PKCθ-deficient B10.PL mice was diminished following overnight activation with anti-CD3 and anti-CD28 (*Figure 1E*). Such Notch activation was CD28-dependent (*Figure 1F*), consistent with an important role of PKCθ downstream of CD28 (*Huang et al., 2002*; *Kong et al., 2011*; *Yokosuka et al., 2008*). As biochemical signaling activity in T cell activation peaks within the first few minutes, we verified that PKCθ-dependent Notch activation can also occur at this time scale. Increased nuclear Notch enrichment could be detected 5 to 20 min after in vitro T cell activation or after injecting mice with 80 µg [4Y] peptide s.c. (*Figure 1G–I*). This effect was corroborated using a second TCR transgenic system, 5C.C7 (*Singleton et al., 2009*) (*Figure 1J*).

## PKC*θ* enhances in vitro Th17 T cell differentiation and T cell proliferation via Notch

To determine functional outcomes of diminished Notch processing in PKCθ-deficient T cells, we analyzed T cell proliferation and differentiation. In accordance with published data (*Keerthivasan et al., 2011*; *Marsland et al., 2004*; *Tan et al., 2006*), T cell proliferation, CD69 and c-Myc upregulation were defective in Tg4^{KO} T cells whereas IL-2 was unaffected (*Figure 2B–D*). Tg4^{KO} T cells showed no defect in Th1 cell differentiation (*Figure 2E–G*). Conversely, the proportion of IL-17A^{+} T cells was significantly reduced under conditions favoring Th17 development (*Figure 2H–J*). Overexpression of NICD at $18.5 \pm 1.5$ fold the level in non-activated primed Tg4^{WT} cells (*Figure 1—figure supplement 1C and D*) so as to constitutively activate Notch in Tg4^{KO} cells led to restoration of T cell proliferation (*Figure 2K*), Th17 cell differentiation and IL-17A secretion (*Figure 2L–N*). Notch1 thus enhances T cell proliferation and differentiation downstream of PKCθ.

## PKC*θ* enhances actin-dependent enrichment of ADAM10 at the T cell: APC interface

We investigated ADAM10 as a key signaling molecule potentially employed by PKCθ for Notch1 activation. We visualized ADAM10 recruitment to the interface between the T cell and the APC by virally transducing Tg4^{WT} and Tg4^{KO} CD4^{+} T cells with ADAM10-GFP. ADAM10-GFP^{+} T cells were imaged as they interacted with H-2^{u} PL8 lymphoma APC presenting the Ac1-9[4Y] antigen. Tg4^{KO} T cells, whether transduced to express ADAM10-GFP or other sensors, formed tight cell couples upon

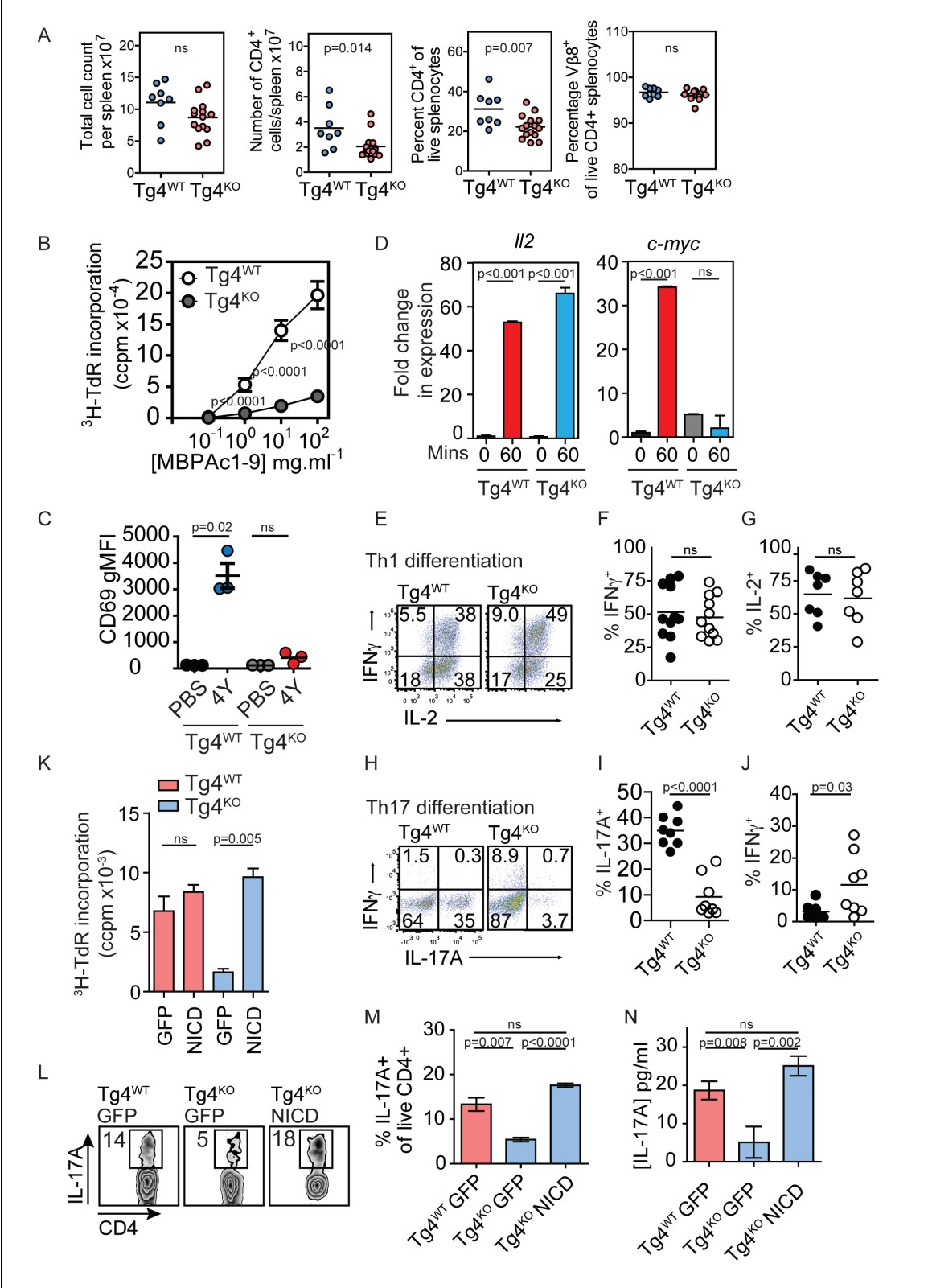

**Figure 2.** Constitutively active Notch rescues defective proliferation, Th17 polarization and IL-17A secretion in PKCθ deficient T cells. (**A**) Splenocytes from Tg4WT and Tg4KO mice were stained for the indicated molecules and absolute cell numbers or percentages are given as indicated. Each data point represents one mouse. Data are combined from mice assayed across three experiments. As previously reported (*Gupta et al., 2008*; *Sun et al., 2000*) in non-TCR transgenic mouse strains, PKCθ deficiency in Tg4 mice resulted in a reduced number and proportion of CD4+ T cells. p values by

*Figure 2 continued on next page*

*Figure 2 continued*

Student's unpaired two-tailed t-test. (B) The proliferation of naïve Tg4[WT] and Tg4[KO] CD4[+] T cells stimulated with irradiated B10.PL splenocytes and a titration of MBPAc1-9[4K] is given. n = 8 Tg4[WT], n = 15 Tg4[KO] mice, each assayed in triplicate for each peptide concentration. Shown is the mean ± SEM. ****p<0.0001 by Student's unpaired two-tailed t-test. (C) Naïve Tg4[WT] and Tg4[KO] mice were injected subcutaneously with 80 μg of MBPAc1-9[4Y] peptide or PBS. After 18 hr splenocytes were immunostained to assess CD69 expression and analyzed by flow cytometry. Gated on live, CD4[+] cells. The mean expression (geometric mean fluorescence intensity, gMFI) ± SEM of CD69 is given. ns p=0.08. One experiment of 2, n = 3 mice per condition. (D) Tg4[WT] or Tg4[KO] mice were injected subcutaneously with 80 μg [4Y] peptide. After 60 min, CD4[+] splenocytes were isolated by MACS, RNA was isolated and the expression of c-myc and IL-2 determined by RT-PCR. n = 3 mice per condition, shown is mean ± SEM. ns = 0.68 by unpaired Student's t-test. (E–G) Splenocytes from Tg4[WT] or Tg4[KO] mice were stimulated with 10 μg/ml [4K] peptide, IL-12 and IL-2 for 7–9 days before restimulation with PMA and ionomycin in the presence of monensin. The proportion of IFNγ and IL-2-producing CD4[+] T cells was determined by intracellular cytokine staining. Shown are representative FACS plots, gated on live, CD4[+] cells and the combined data from all replicates shown as mean ± SEM n = 7–11 independent biological replicates ns = 0.71 (F) and 0.73 (G) by unpaired Student's t-test. (I–J) Splenocytes from Tg4[WT] or Tg4[KO] mice were stimulated with 10 μg/ml [4K] peptide, IL-6, IL-1β, IL-23, anti-IFNγ and anti-IL-4 for 7–9 days before restimulation with PMA and ionomycin in the presence of monensin. The proportion of IFNγ and IL-17A-producing CD4[+] T cells was determined by intracellular cytokine staining. Shown are representative FACS plots, gated on live, CD4[+] cells and the combined data from all replicates shown as mean ± SEM. n = 8 independent biological replicates, p<0.0001 (I) p=0.03 (J) by Student's t-test. (K) Splenocytes from Tg4[WT] and Tg4[KO] mice were stimulated with [4K] peptide and IL-2 before transduction with a retrovirus encoding NICD and GFP or GFP alone. After 72 hr the incorporation of [3]H thymidine was measured. n = 3 replicate transductions per condition, mean values ± SEM. *** = 0.0005, ns = 0.31. One representative experiment of four. (L, M) Splenocytes from Tg4[WT] and Tg4[KO] mice were stimulated with [4K] peptide in the presence of IL-6, IL-1β, TGFβ and IL-23 before transduction with a retrovirus encoding NICD and GFP or GFP alone. After 96 hr of further culture with IL-6, IL-1β, TGFβ and IL-23 the cells were restimulated with PMA and ionomycin in the presence of monensin before intracellular staining for the expression of IL-17A. (L) shows representative FACS data. The mean ± SEM of three replicates from one experiment of four is shown in M. ns = 0.06 by Student's t-test. (N) The mean concentration of IL-17A was measured in supernatants from triplicate cultures of Tg4[WT] and Tg4[KO] cells T cells transduced with NICD or GFP alone under Th17-polarising conditions. p values by Student's t-test. One representative experiment of three.

The following figure supplement is available for figure 2:

**Figure supplement 1.** Tg4[KO] mice display largely unperturbed immune cell distributions.

APC contact albeit with a slightly reduced frequency compared to Tg4[WT] T cells (30 ± 5% versus 50 ± 4%, p=0.02) with comparable gross T cell morphology, as characterized in the next paragraph. Such effective cell coupling allowed an analysis of the interface recruitment of GFP-tagged signaling intermediates and the spatiotemporal patterns thereof. ADAM10-GFP was recruited rapidly and transiently to the interface of Tg4[WT] T cells and APC (*Figure 3A,B*; *Figure 3—figure supplement 1A*, *Video 1*) consistent with previous work in AND T cells (*Guy et al., 2013*). In contrast, ADAM10-GFP was not enriched at the interface of Tg4[KO] T cells (*Figure 3A,B*; *Figure 3—figure supplement 1A*, *Video 2*). In Tg4[WT] cells, ADAM10 was enriched in the interface lamellum, an actin-based signaling structure (*Roybal et al., 2015b*). Impairment of lamellum formation with 40 nM Jasplakinolide (*Figure 3—figure supplement 1B*) (*Roybal et al., 2015a*) prevented ADAM10 interface recruitment (*Figure 3A,B*; *Figure 3—figure supplement 1A*) and Notch cleavage following stimulation with anti-CD3/28 (*Figure 3C*). The defect in lamellal ADAM10 recruitment upon PKCθ-deficiency was selective since the lamellal accumulation of Themis, a protein with substantially more prominent and persistent lamellal accumulation than ADAM10, was only moderately impaired (*Figure 3—figure supplement 1C–E*). Together, these data suggest that actin-dependent ADAM10 recruitment to the T cell:APC interface at the early peak of T cell signaling activity mediates efficient Notch1 activation downstream of PKCθ. In AND T cells strong stimuli cause concerted accumulation of the TCR, Vav and Notch at the T cell/APC interface as related to efficient Notch activation (*Guy et al., 2013*). Actin dynamics may thus mediate coordinated interface accumulation of both ADAM10 and Notch. It needs to be determined how PKCθ and Vav-dependent actin dynamics are related.

Previous work has linked PKCθ to actin regulation (*Sasahara et al., 2002*; *Sims et al., 2007*; *Villalba et al., 2002*). By visualizing actin dynamics in Tg4[WT] and Tg4[KO] T cells with F-tractin-GFP (*Johnson and Schell, 2009*) (*Videos 3* and *4*), multiple elements of the actin-dependent establishment of a tight T cell:APC interface were modestly impaired in cells lacking PKCθ. The spreading of F-actin to the periphery of the interface was delayed in Tg4[KO] T cells (*Figure 3D,E*; *Figure 3—figure supplement 2A*). The interface diameter of Tg4[KO] T cells was significantly (p<0.05) reduced across multiple time points (*Figure 3F,G*), as confirmed by electron microscopy (*Figure 3—figure*

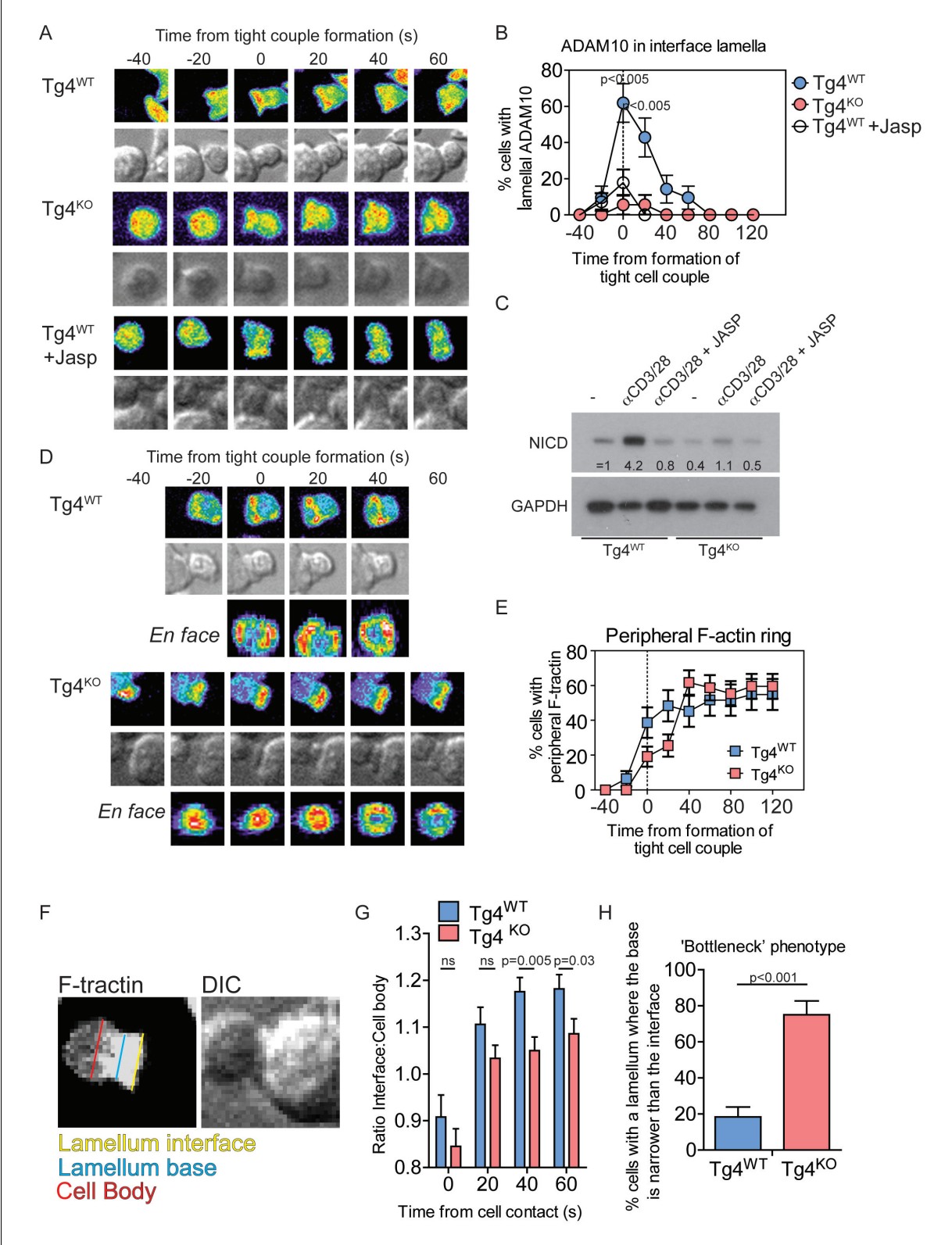

**Figure 3.** PKCθ mediates transient actin-dependent recruitment of ADAM10 to the T cell lamellum. (**A**) Tg4[WT] CD4[+] T cells, treated with 40 nM Jasplakinolide (bottom) or not (top), and Tg4[KO] (middle) CD4[+] T cells expressing ADAM10-GFP were activated with PL8 cells presenting the Ac1-9[4Y] peptide. Given are representative images showing pseudocolored (purple to red) maximum projections of the ADAM10-GFP fluorescence and a reference DIC bright field image at times relative to the formation of a tight couple between T cell and APC. The entire image sequences are given in *Figure 3 continued on next page*

*Figure 3 continued*

*Video 1* (Tg4[WT]) and 2 (Tg4[KO]). (B) The graph shows the percentage of T cells with lamellal accumulation of ADAM10-GFP at the time relative to couple formation ± SEM. Differences in lamellal accumulation between Tg4[WT] and Tg4[KO] and Jasplakinolide-treated Tg4[WT] T cells at time points 0:00 and 0:20 were each significant with $p \leq 0.005$ (Tg4[KO] versus Tg4[WT] 0:00 p=0.001, 0:20 p=0.005; Tg4[WT] +Jasp versus Tg4[WT] 0:00 p=0.004, 0:20 p=0.001, by proportions z-test). 18–28 cell couples were analyzed per condition (57 total). Full pattern analysis is given in *Figure 3—figure supplement 1A*. (C) CD4[+] blasts from Tg4[WT] and Tg4[KO] mice (four days after stimulation) were restimulated for 18 hr with anti-CD3 and anti-CD28 ±40 nM Jasplakinolide or left unstimulated as indicated. NICD and GAPDH expression in protein extracts was measured by Western blotting. One representative experiment of three. (D) Tg4[WT] and Tg4[KO] CD4[+] T cells expressing F-tractin-GFP were activated with PL8 cells presenting the Ac1-9[4Y] peptide. Representative images are given as in A. The entire image sequences are given in *Video 3* (Tg4[WT]) and 4 (Tg4[KO]). (E) The percentage of cell couples with predominantly peripheral F-tractin-GFP accumulation is given as in B. The difference in peripheral accumulation between Tg4[WT] and Tg4[KO] T cells at joint time points 0:00 and 0:20 was significant (p=0.01 by proportions z-test, 31, 47 cell couples were analyzed per condition). Full pattern analysis is given in *Figure 3—figure supplement 2A*. (F) An example image of a T cell exhibiting the 'bottleneck' phenotype, defined as having a diameter minimum between the interface and the widest part of the cell body, is given as a grey scale F-tractin-GFP maximum projection together with a matching DIC bright field image. Measurement positions to determine the interface width (yellow) relative to the cell body (red) or the presence of a necking phenotype (blue) are shown. (G) The relative interface diameter was determined by relating the interface diameter to the widest part of the cell body and is given relative to the time of tight cell coupling. Shown is the mean ratio ± SEM. ns p=0.33 (0 s) and 0.149 (20 s) by unpaired, two-tailed Student's t-test. 49 (WT) 35 (KO) cell couples were analyzed per condition. (H) The percentage of T cells displaying a bottleneck phenotype in at least one timepoint during the first 60 s after coupling is given. ***p<0.001 by proportions z-test. 35 cell couples were analyzed per condition.

The following figure supplements are available for figure 3:

**Figure supplement 1.** PKCθ enables transient recruitment of ADAM10 to the T cell lamellum.

**Figure supplement 2.** PKCθ enhances interface actin dynamics.

supplement 2B). The lamellum connecting the T cell body to the interface was smaller in Tg4[KO] T cells as it showed a significantly (p<0.001) larger constriction or 'neck' (*Figure 3F,H*). Long lamella (>2.5 µm) did not occur (*Figure 3—figure supplement 2C*). Tg4[KO] T cells thus displayed modest defects across multiple elements of actin-driven cell spreading consistent with the slightly reduced efficiency of tight cell coupling.

## PKC*θ* phosphorylates and localizes Coronin1A

We sought to identify actin regulators mediating the modest actin modulation by PKCθ. Coronin1A inhibits the Arp2/3 complex (*Humphries et al., 2002*; *Oku et al., 2005*) and regulates clearance of actin from the NK immune synapse (*Mace and Orange, 2014*). Furthermore, Coronin is an established interactor with and substrate of PKC (*Cai et al., 2005*; *Siegmund et al., 2015*). Coronin1A was highly enriched at the interface of Tg4[WT] and Tg4[KO] T cells (*Figure 4A,B*; *Figure 4—figure supplement 1A*; *Videos 5* and *6*). Similar to actin, Coronin1A spreading to the interface periphery was delayed in Tg4[KO] cells, leaving substantially more Coronin 1A in the lamellum. Given that Coronin 1A is a negative regulator of actin dynamics its enhanced enrichment in the lamellum is consistent with the concurrent, localized impairments in actin, T cell morphology and ADAM10 recruitment. As a specificity control, the spatiotemporal distribution of the dominant F-actin severing protein Cofilin (*Roybal et al., 2016*; *Singleton et al., 2011*) was unaffected by

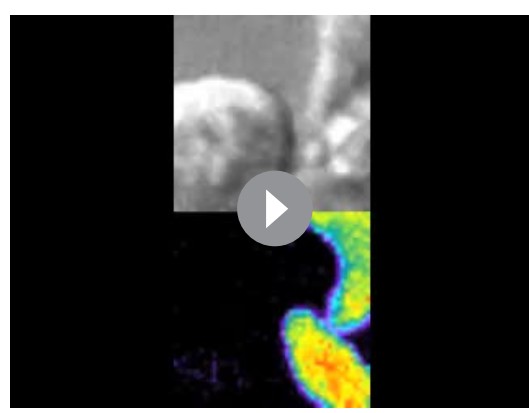

**Video 1.** ADAM10-GFP accumulates rapidly and transiently at the interface between Tg4[WT] CD4[+] T cells and PL8 APCs. A representative interaction of a Tg4[WT] CD4[+] T cell expressing ADAM10-GFP with a PL8 APC presenting the Ac1-9[4Y] peptide is shown. Top: DIC images. Bottom: Top-down maximum projections of 3D fluorescence data are shown in a rainbow-like, false-color intensity scale (increasing from blue to red). 20 s intervals in video acquisition are played back as two frames per second. Tight cell coupling occurs in frame 3 (1 s indicated video time).

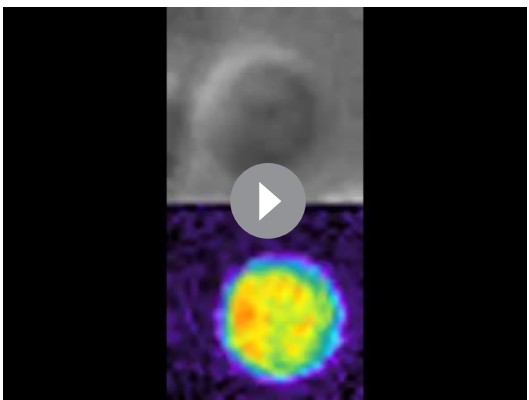
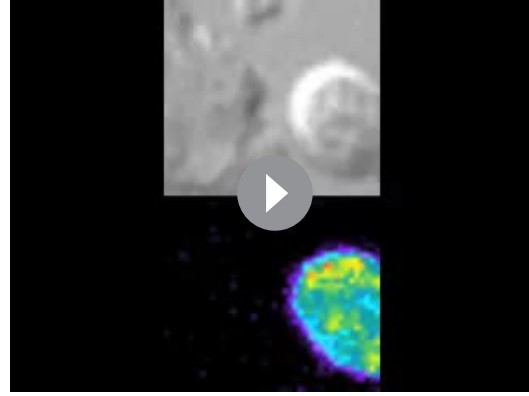

**Video 2.** ADAM10-GFP does not accumulate at the interface between Tg4[KO] CD4[+] T cells and PL8 APCs. A representative interaction of a Tg4[KO] CD4[+] T cell expressing ADAM10-GFP with a PL8 APC presenting the Ac1-9[4Y] peptide is shown as in *Video 1*. Tight cell coupling occurs in frame 5 (2 s indicated video time).

**Video 3.** F-tractin-GFP accumulates rapidly at the interface between Tg4[WT] CD4[+] T cells and PL8 APCs. A representative interaction of a Tg4[WT] CD4[+] T cell expressing F-tractin-GFP with a PL8 APC presenting the Ac1-9[4Y] peptide is show as in *Video 1*. Tight cell coupling occurs in frame 6 (4 s indicated video time). Immediate spreading of the majority of F-actin to the edge of the interface is visible.

PKCθ deficiency (*Figure 4—figure supplement 1A*).

Next, we investigated phosphorylation of Coronin 1A by PKCθ in Tg4 T cells. Coronin activity is negatively regulated by serine/threonine phosphorylation, which can be induced by phorbol ester treatment (*Cai et al., 2005*; *Oku et al., 2008*, *2012*). To allow detection of changes in Coronin1A phosphorylation, we prevented Coronin1A dephosphorylation by treating cells with the phosphatase inhibitor Calyculin A (*Oku et al., 2008*, *2012*). Treating Tg4[WT] T cells with PMA and Calyculin A resulted in a shift in the ratio of phosphorylated to non-phosphorylated Coronin1A from $0.3 \pm 0.1$ to $2.8 \pm 1.1$ fold, indicative of efficient Coronin1A phosphorylation (*Figure 4C,D*). This shift was significantly ($p<0.05$) smaller in Tg4[KO] cells $(0.25 \pm 0.05$ to $0.8 \pm 0.25$ fold) (*Figure 4C,D*) demonstrating that PKCθ is required for efficient PMA-induced phosphorylation of Coronin1A. Together, our data suggest (*Figure 4E*) that in Tg4[WT] T cells PKCθ (*Figure 4—figure supplement 1B*; *Video 7*) inactivates Coronin 1A selectively in the region of most intense stimulating signaling, i.e. the center of the T cell:APC interface with effects extending across the entire lamellum but not reaching the peripheral actin ring. Thus PKCθ locally inhibits Coronin 1A-mediated attenuation of actin dynamics, promoting the formation of a strong actin-based lamellum. With regard to Notch1 activation this allows for the efficient actin-driven recruitment of ADAM10 to the T cell:APC interface. This mechanism of enhanced Notch processing peaks within the first few minutes of T cell activation. Such early signaling emphasis is consistently observed in Tg4[WT] cells (*Figure 4—figure supplement 1B*) and other TCR transgenic systems, where it extends to the nuclear localization of other transcription factors, NFAT and NFκB (*Roybal et al., 2015b*; *Singleton et al., 2009*). We have thus

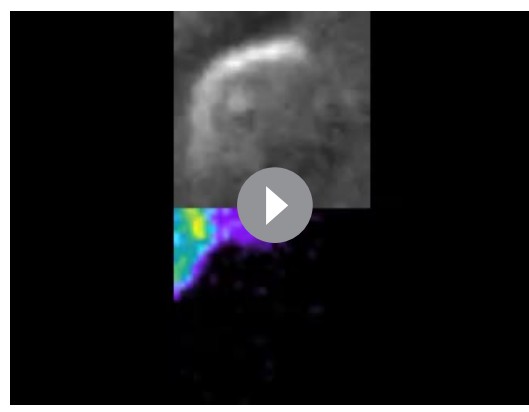

**Video 4.** F-tractin-GFP accumulates rapidly at the interface between Tg4[KO] CD4[+] T cells and PL8 APCs. A representative interaction of a Tg4[KO] CD4[+] T cell expressing F-tractin-GFP with a PL8 APC presenting the Ac1-9[4Y] peptide is shown as in *Video 1*. Tight cell coupling occurs in frame 4 (2 s indicated video time). Delayed spreading of the majority of F-actin to the edge of the interface is visible.

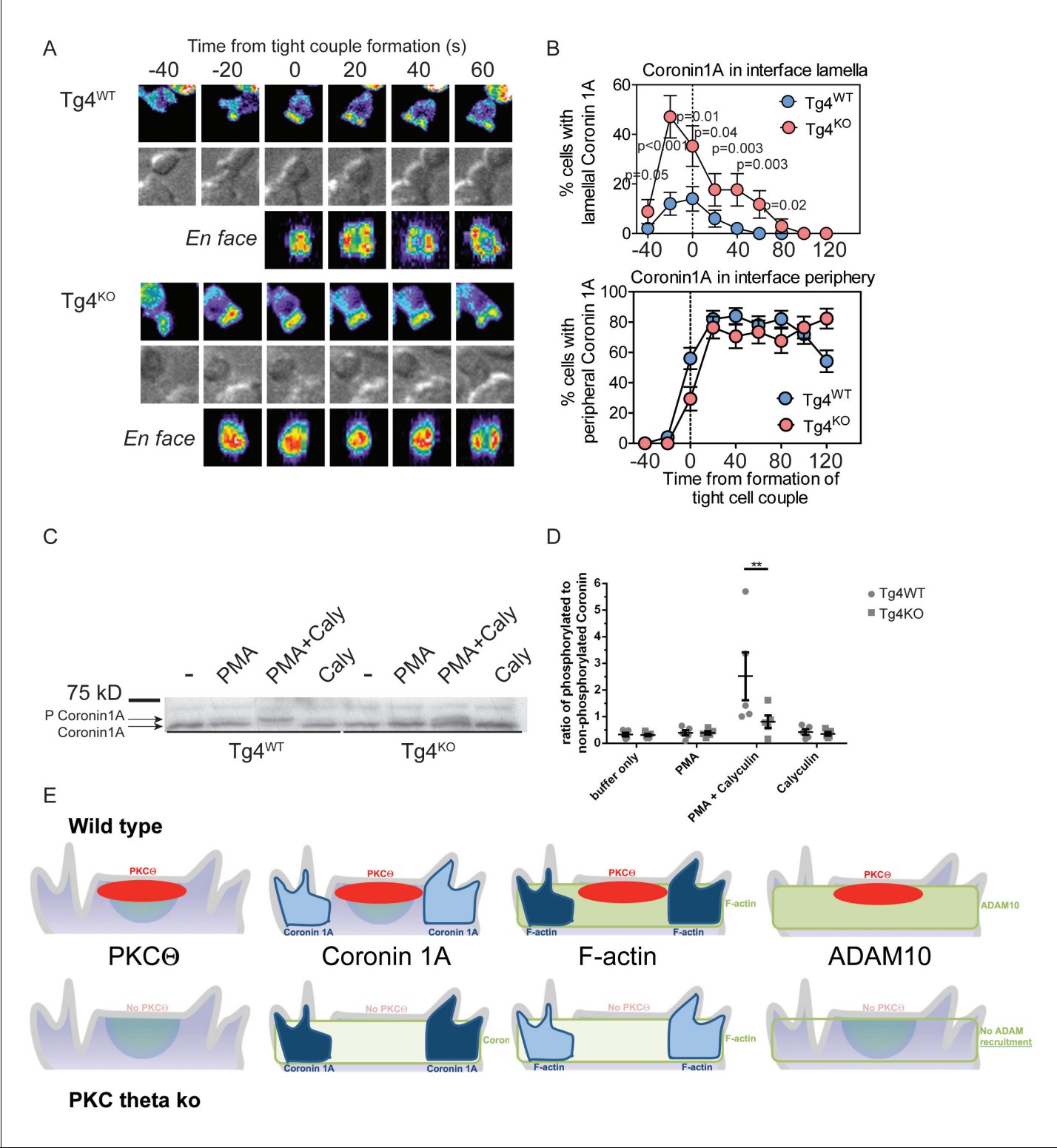

**Figure 4.** PKCθ phosphorylates and localizes Coronin1A. (**A**) Tg4^WT and Tg4^KO T cell expressing coronin1A-GFP were activated with PL8 cells presenting the Ac1-9[4Y] peptide. Representative images are given as in *Figure 3A*. The entire image sequences are given in *Video 5* (Tg4^WT) and 6 (Tg4^KO). (**B**) The percentage of cell couples with predominantly lamellal (top) and peripheral (bottom) Coronin1A-GFP accumulation is given as in *Figure 3B*. The differences in lamellal accumulation between Tg4^WT and Tg4^KO T cells at time points −40 to 80 were each significant with p≤0.05 by proportions z-test. 50, 34 cell couples were analyzed per condition. Full pattern analysis is given in *Figure 4—figure supplement 1A*. (**C**) Shown is a representative Phos-tag western blot of protein extracts from Tg4^WT or Tg4^KO T cells stimulated with PMA and/or Calyculin A (Caly) for 5 min as probed

*Figure 4 continued on next page*

*Figure 4 continued*

with anti-coronin1A. (D) Given is the quantification of four independent experiment as in C as the mean ratio of the top (phospho) and lower (non-phospho) Coronin 1A bands ± SEM. * indicates p<0.05 Tg4$^{WT}$ versus Tg4$^{KO}$ T cells by two-way ANOVA with Sidak's correction for multiple comparisons. (E) A graphical summary of the proposed mechanism of the enhancement of Notch activation by PKCθ is given. The top and bottom rows illustrate Tg4$^{WT}$ or Tg4$^{KO}$ T cells, respectively. Each individual panel shows the interface part of the T cell that contacts the APC (not shown on top). Separate panels are drawn from left to right for PKCθ (as also included in the other panels), Coronin1A, F-actin and ADAM10. Colors denote preferential accumulation patterns, central (red), lamellal (green) and peripheral (blue). Shade of color denotes the extent of accumulation.

The following figure supplement is available for figure 4:

**Figure supplement 1.** Interface recruitment of signaling intermediates peaks within the first three minutes in Tg4$^{KO}$ CD4$^+$ T cells.

identified a spatially restricted, actin-dependent mechanism of Notch activation downstream of PKCθ (*Figure 4E*). Future work will determine how the enhancement of Notch activation by PKCθ is integrated with PKCθ-dependent NFκB activation (*Gruber et al., 2009*; *Sun et al., 2000*) in the regulation of T cell differentiation.

A key feature of our mechanism of PKCθ function is that it connects signaling at the time scale of minutes to outcomes in cellular differentiation over days. While causally connecting such divergent time scales is a great challenge, there is precedent. 15 min of contact between a primed T cell and a professional APC is sufficient to trigger T cell proliferation 24 hr later (*Iezzi et al., 1998*). Similarly, 1 hr of ZAP-70 activity can trigger substantial negative selection (*Au-Yeung et al., 2014*). Differential signaling kinetics may also regulate Treg induction (*Miskov-Zivanov et al., 2013*). On an even shorter time scale 5 min of TGFβ incubation saturates Smad2 phosphorylation at 1 hr (*Vizán et al., 2013*). In T cell activation, it has been argued that a time delay in the onset of activating versus inhibitory signaling from 2 to more than 5 min, respectively, may play an important role in the induction of anergy in response to high doses of antigen (*Wolchinsky et al., 2014*). In B cells a single pulse of BCR engagement can trigger the nuclear accumulation of NFκB for 6 hr (*Damdinsuren et al., 2010*). While mechanisms linking rapid proximal signaling to later cell function largely remain to be determined, the well-supported existence of such causal links is consistent with our model of PKCθ-dependent Notch activation.

## Materials and methods

### Mice

All mice were maintained under SPF conditions with *ad libitum* access to water and standard chow at the University of Bristol. All animal experiments were carried out under the UK Home Office Project Licence number 30/2705 held by David Wraith and the study was approved by the University of Bristol ethical review committee. B10.PL, 5C.C7 (*Seder et al., 1992*) and Tg4 (*Liu et al., 1995*) mice were bred in-house at the University of Bristol. PKCθ-deficient Tg4 mice were generated by cross-breeding Tg4 mice with C57BL/6 *prkcq*$^{-/-}$ mice (a gift of A. Poole, University of Bristol, originally generated by D. Littman (*Sun et al., 2000*) for >8 generations. The genetic status of each animal was assessed by PCR as previously described (*Sun et al., 2000*). B10.PL PKCθ KO mice were obtained by breeding Tg4$^{KO}$ mice with B10.PL mice.

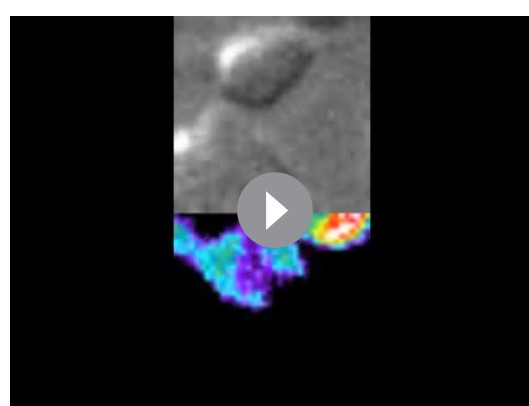

**Video 5.** Coronin1A-GFP accumulates rapidly at the interface between Tg4$^{WT}$ CD4$^+$ T cells and PL8 APCs. A representative interaction of a Tg4$^{WT}$ CD4$^+$ T cell expressing Coronin1A-GFP with a PL8 APC presenting the Ac1-9[4Y] peptide is shown as in *Video 1*. Tight cell coupling occurs in frame 4 (2 s indicated video time). Immediate spreading of the majority of Coronin1A-GFP to the edge of the interface is visible.

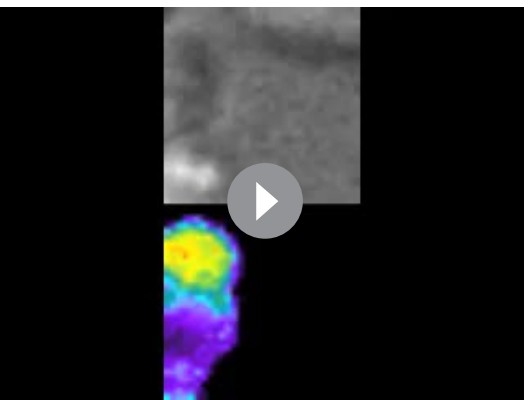

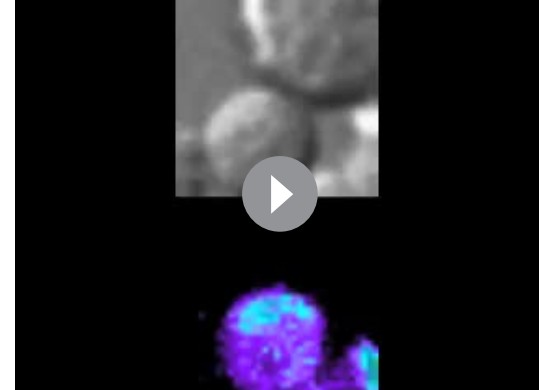

**Video 6.** Coronin1A-GFP accumulates rapidly at the interface between Tg4[KO] CD4[+] T cells and PL8 APCs. A representative interaction of a Tg4[KO] CD4[+] T cell expressing Coronin1A-GFP with a PL8 APC presenting the Ac1-9[4Y] peptide is shown as in *Video 1*. Tight cell coupling occurs in frame 3 (1 s indicated video time). Transient lamellal accumulation of the majority of Coronin1A-GFP is visible.

**Video 7.** PKCθ-GFP accumulates at the center of the interface between Tg4[WT] CD4[+] T cells and PL8 APCs. A representative interaction of a Tg4[WT] CD4[+] T cell expressing PKCθ-GFP with a PL8 APC presenting the Ac1-9[4Y] peptide is shown as in *Video 1*. Tight cell coupling occurs in frame 4 (2 s indicated video time). Central accumulation of the majority of PKCθ-GFP is visible.

## Cell lines

The PL8 H-2[u] expressing, antigen-presenting B cell lymphoma was prepared in our laboratory (*Wraith et al., 1992*). The H-2[k] expressing CH27 B cell lymphoma was prepared as described previously (*Haughton et al., 1986*) and obtained from Mark Davis, Stanford University. Both cell lines proved mycoplasma free by PCR and were validated by staining for MHC class II expression and assessing their ability to present antigen to relevant T cell lines.

## Cell culture

Lymphoid tissue was dissociated using standard protocols and red blood cells removed using Red Cell Lysis buffer (Sigma). Unless otherwise stated, cells were cultured in complete RPMI 1640 (Lonza; supplemented with 25 mM HEPES, 50 U/ml Pen/Strep, 2 mM L-Glutamine and 50 µM 2-mercaptoethanol) with 5–10% FCS (BioSera, Hyclone). PL8 and CH27 cells were maintained in complete RPMI with 10% FCS. Th17 cells were generated and maintained in IMDM (Lonza; supplemented with 50 U/ml Pen/Strep, 2 mM L-Glutamine and 50 µM 2-mercaptoethanol) containing 10% FCS.

## T cell isolation and stimulation

For isolation of double negative thymocytes, thymi were gently disaggregated on ice in 5% FCS/PBS. Cells were stained at 4°C with CD4-FITC and CD8a-APC antibodies. Propidium iodide was added immediately prior to flow cytometric sorting of viable CD4-CD8a- thymocytes using a BD Influx cell sorter. For in vitro stimulation experiments, naïve CD4[+] T cells were isolated from spleen and axillary, brachial and inguinal lymph nodes using either EasySep Mouse Naïve CD4[+] T cell isolation kit (Stem Cell Technologies) or MagniSort Mouse Naïve CD4[+] T cell enrichment kit (eBioscience) according to the manufacturers' instructions. For ex-vivo analysis of activated T cells, CD4[+] cells were enriched with Mouse CD4[+] T cell enrichment kit II (Miltenyi Biotech) according to the manufacturer's instructions. Naïve or pre-activated CD4[+] T cell cells were stimulated with plate-bound anti-CD3 (2C11, eBioscience or BioExcel, 1 µg/ml or as indicated) and anti-CD28 (37.51, eBioscience or BioExcel, 2 µg/ml). Alternatively, cells were stimulated with PL8 cells and MBPAc1-9 [4K] or [4Y] peptide (GL Biochem) or CH27 cells and MCC 88–103 peptide at the concentration indicated. Where indicated, cells were incubated with PMA (Sigma, 20 ng/ml), Calyculin A (Sigma, 100 nM) or Jasplakinolide (Tocris, 40 nM).

## T helper differentiation

Suspensions of splenocytes from Tg4$^{WT}$ and Tg4$^{KO}$ mice were stimulated with 10 µg/ml MBP Ac1-9 [4K] peptide. For T$_H$1 generation, cells were cultured in complete RPMI containing 10 ng/ml IL-12 (Peprotech) and 20 U/ml rhIL-2 (R and D systems). For T$_H$17 cells, culture was performed in complete IMDM containing 25 ng/ml IL-6, 10 ng/ml IL-1$\beta$, 2 ng/ml TGF$\beta$ (all Peprotech), 10 ng/ml IL-23 (eBioscience), 50 µg/ml anti-IFN$\gamma$ (XMG1; BioExcel) and 10 µg/ml anti-IL-4 (11B11; BioExcel).

## Western blotting

CD4$^+$ T cells were stimulated as indicated and washed with ice-cold PBS before protein was extracted in RIPA buffer supplemented with protease and phosphatase inhibitor cocktails (all from Pierce) ($1-2 \times 10^7$ cells/ml). Lysates were centrifuged for 10 min at 17,000xg and the soluble fraction denatured in Laemmli buffer before resolution by SDS-PAGE on 4–12% gels (NuPAGE), transfer to PVDF membrane and immunodetection using standard ECL protocols. Where indicated, samples were resolved on 12.5% gels supplemented with 50 µM PhosTag reagent (Wako) and 100 µM ZnCl$_2$. For PhosTag experiments, cells were washed with HBSS instead of PBS. The following antibodies were used Notch1; (D1E11, Cell Signaling), GAPDH (D16H11, Cell Signaling), c-myc (E910, Santa Cruz Biotechnology), Coronin1A (H300, Santa Cruz Biotechnology), anti-rabbit and anti-mouse HRP conjugates (Sigma).

## Flow cytometry and cytokine measurements

Non-viable cells were excluded from all analyses using Live/Dead eF780 dye (1:1000, eBioscience). Surface staining was performed in PBS containing 0.5% FCS and 2 mM EDTA. Intracellular cytokine staining (ICCS) was performed on cells stimulated with PMA (10 ng/ml) and ionomycin (1 µg/ml) in the presence of GolgiStop (BD Bioscience, 1:1000) for 4 hr. Cells were surface stained before fixation and permeabilization using eBioscience reagents. Staining for intracellular Notch1 and FoxP3 was performed after fixation and permeabilization using FoxP3 staining kit reagents (eBioscience). The following antibodies, all purchased from eBioscience and/or Biolegend, were used; CD4 Alexa700 (GK1.5, 1:100), CD69 FITC (H1.2F3, 1:100), Notch1-PE (mN1A, 1:100, Biolegend), CD8a APC (53–6.7, 1:200), CD19 (1D3, 1:200), B220 FITC (RA3-6B2, 1:100), Vb8.1/2 FITC (KJ16-133, 1:100), FoxP3 PE (FJK-16S, 1:100, eBioscience), CD25 PE-Cy7 (PC61.5, 1:300), IFN$\gamma$ PE-Cy7 (XMG1, 1:400), IL-2 eF450 (JES6-5H4, 1:100) and IL-17A PE or PE-Cy7 (17B7, 1:2–400).

Soluble IL-17A was detected in culture supernatant by ELISA using Ready-Set-Go ELISA kit (eBioscience).

## T cell transduction

The cDNA encoding the IC domain of murine Notch1 was obtained from Addgene (plasmid number 20183). The IC domain was amplified by PCR with the primers ACCGCGGTGGCGGCCATGCAGCA TGGCCAGCTCT and CGGGCTAGAGCGGCCTTATTTAAATGCCTCTGGAATGT and cloned into the Not1 site of pGC-IRES-GFP (*Costa et al., 2000*) using In-Fusion HD reagents (Clontech). cDNA encoding murine Adam10, obtained from Sinobiological (Genbank number NM_007399.3), was amplified with primers ACCGCGGTGGAGGCCAAGATGGTGTTGCCGACAGT and GGCGACCGG TGGATCTCCACCGCGTCGCATGTGTCCCATT and cloned into BamH1 and Not1 sites of pGC-GFP using In-Fusion reagents such that the C-terminus of ADAM10 was fused to GFP. The pGC-IRES-GFP vector was used as a negative control. The constructs used to express other sensors including Coronin1A-GFP, Cofilin-GFP, Themis-GFP, F-Tractin-GFP and PKC$\theta$–GFP have been previously described (Table 1 in [*Roybal et al., 2015b*] and [*Roybal et al., 2016*]). Retrovirus was generated by transfecting Phoenix-E cells using calcium phosphate precipitation. For imaging experiments, Tg4 T cells were infected by centrifugation with viral supernatant 24 hr after stimulation with 10 µg.ml$^{-1}$ [4K] and 20 U.ml$^{-1}$ rhIL-2. Immediately following transduction culture media was replaced with complete RPMI supplemented with 20–40 U/ml rhIL-2. For T$_H$17 Notch rescue experiments, Tg4$^{WT}$ and Tg4$^{KO}$ splenocytes were cultured for 24 hr under T$_H$17-polarising conditions (as described above) before transduction. Immediately following transduction, the culture medium was replaced with complete IMDM containing 20 ng/ml IL-23 and 50 µg/ml anti-IFN$\gamma$. T cells were analyzed by imaging or flow cytometry 4–5 days after transduction.

## Proliferation measurements

72 hr following transduction, 2.5 µCi $^3$H thymidine/ml (Perkin Elmer) was added to culture wells. After 16 hr incubation, incorporation of $^3$H thymidine was measured by scintillation counting.

## RT-PCR

CD4$^+$ T cells were stimulated and isolated as indicated in figure legends and RNA was extracted using either RNeasy Mini kit (Qiagen) or TRI Reagent (Sigma Aldrich). cDNA was generated using Superscript III polymerase (Life Technologies) and real time PCR performed using a SYBR green PCR Mastermix (Life Technologies). Primers; *il2* sense; AGCAGCTGTTGATGGACCTA, *il2* antisense; CGCAGAGGTCCAAGTTCAT, *cmyc* sense; TTGAAGGCTGGATTTCCTTTGGGC, *cmyc* antisense; TCGTCGCAGATGAAATAGGGCTGT, *Hes1* sense; AAAGATAGCTCCCGGCATTC, *Hes1* antisense; TGCTTCACAGTCATTTCCAGA, *β2M* sense; GCTATCCAGAAAACCCCTCAA, *β2M* antisense; CGGGTGGAACTGTGTTACGT. Data were analysed using the $2^{-\triangle\triangle CT}$ method, normalized to β2microglobulin.

## Live cell imaging

Live cell imaging was performed as described in detail before (*Singleton et al., 2009*). Using FACS, GFP$^+$ transductants were sorted to a five-fold range of expression around 2 µM, the lowest concentration visible by microscopy and often within the range of endogenous protein amounts (*Roybal et al., 2016*). PL8 cells were pre-loaded with 10 µg/ml [4Y] for >4 hr and combined with pre-sorted GFP$^+$ Tg4 T cells in a glass-bottom plate on the stage of a spinning disk microscope system (UltraVIEW 6FE system, Perkin Elmer; DMI6000 microscope, Leica; CSU22 spinning disk, Yokogawa). GFP data were collected as 21 z-sections at 1 µm intervals every 20 s. All imaging was performed at 37$^0$C in PBS containing 10% FCS, 1 mM CaCl$_2$ and 0.5 mM MgCl$_2$. Images were exported in TIFF format and analyzed with the Metamorph software (Molecular Devices). Cell couples were identified using the differential interference contrast (DIC) bright field images. The subcellular localization of GFP-tagged protein sensors at each time point was classified into one of six previously defined stereotypical patterns (*Singleton et al., 2009*) that reflect cell biological structures driving signaling organization (*Roybal et al., 2013*). Briefly, interface enrichment of fluorescent proteins at less than 35% of the cellular background was classified as no accumulation. For enrichment above 35% the six, mutually exclusive interface patterns were: accumulation in a large protein complex at the center of the T cell:APC interface (central), accumulation in a large T cell invagination (invagination), accumulation that covered the cell cortex across central and peripheral regions (diffuse), accumulation in a broad actin-based interface lamellum (lamellum), accumulation at the periphery of the interface (peripheral) or in smaller membrane protrusions (asymmetric).

## Immunofluorescence staining

Pre-activated Tg4$^{WT}$ and Tg4$^{KO}$ CD4$^+$ T cells (4 days after activation) were combined with PL8 APC pre-incubated with 10 µM MBPAc1-9[4Y] for 15 min before fixation with 4% PFA. Alternatively, Tg4 or 5C.C7 T cells were activated in vivo by s.c. injection with 80 µg MBPAc1-9[4Y] or MCC (88-103) respectively before cell isolation and fixation. Following permeabilization with 0.05% Triton X-100 cells were immunolabelled with anti-Notch1 IC domain (D1E11, Cell Signaling) with an anti-rabbit Alexa488-conjugated secondary antibody (Life Technologies) and counterstained with DAPI and Phalloidin Alexa647 (Life Technologies). Alternatively, cell couples were stained with anti-Notch Alexa647 (Abcam, ab194122) and anti-CD4 FITC. Images were acquired on a Leica SP5 confocal microscope and image analysis was performed in Metamorph and Volocity (Perkin Elmer).

## Electron microscopy

Electron microscopy experiments were executed as described in detail in *Roybal et al. (2015b)*. Briefly, Tg4$^{WT}$ or Tg4$^{KO}$ CD4$^+$ T cells and peptide-loaded PL8s were centrifuged together for 30 s at 350 g to synchronize cell coupling, the cell pellet was immediately resuspended to minimize unspecific cell coupling and cellular deformation and the cell suspension was further incubated at 37 degree C. After 2 and 5 min for early and late time points, respectively, the cell suspension was high pressure frozen and freeze substituted to Epon. Ultrathin sections were analyzed in an FEI Tecnai12 BioTwin equipped with a bottom-mount 4*4K EAGLE CCD camera. T cell:APC couples were

identified in electron micrographs through their wide cellular interface. As described in detail in *Roybal et al. (2015b)*, the time point assignment of cell couples was filtered with morphological criteria post acquisition using the presence of a uropod and T cell elongation.

## Statistical methods

No statistical methods were used to predetermine the sample size. The significance of pairwise comparisons was measured by Student's t-test. Where multiple comparisons were made, significance was determined by ANOVA with Tukey correction. The statistical significance in differences in percentage occurrence was calculated with a proportions z-test.

## Acknowledgements

GJB was supported by the Wellcome Trust Dynamic Cell Biology programme grant 086779/Z/08/A RA, HMT, KEM, LAH were supported by the Wellcome Trust Dynamic Cell Biology programme grant 102387/Z/13/Z DJC was supported by a University of Bristol PhD studentship CSP was supported by a senior research fellowship from the Multiple Sclerosis Society CW was supported by the ERC grant PCIG-GA-2012–321554 DW was supported by a Wellcome Trust Programme Grant 091074/Z/09/Z We acknowledge the MRC, the Wolfson Foundation and the University of Bristol for supporting the Wolfson Bioimaging Facility. We thank Alan Leard and Dr Katy Jepson for microscopy support, Dr Andrew Herman and the University of Bristol Flow Cytometry Facility for cell sorting and analysis and Ella Sheppard, Louise Falk and Anna Lewis for technical support.

## Additional information

### Funding

| Funder | Grant reference number | Author |
| --- | --- | --- |
| Wellcome Trust | 086779/Z/08/A | Graham J Britton |
| Wellcome Trust | 102387/Z/13/Z | Rachel Ambler<br>Helen M Tunbridge<br>Kerrie E McNally<br>Lea A Hampton-O'Neil |
| University of Bristol | PhD studentship | Danielle J Clark |
| Wellcome Trust | 091074/Z/09/Z | Elaine V Hill<br>David Cameron Wraith |
| Multiple Sclerosis Society | 900/08 | Catherine Sabatos-Peyton |
| European Research Council | PCIG-GA-2012-321554 | Christoph Wülfing |

The funders had no role in study design, data collection and interpretation, or the decision to submit the work for publication.

### Author contributions

GJB, Conceptualization, Data curation, Formal analysis, Investigation, Writing—original draft, Writing—review and editing; RA, EVH, HMT, KEM, BRB, PB, LAH-O'N, Investigation; DJC, Methodology; CS-P, Conceptualization, Supervision; PV, Investigation, Methodology; CW, Conceptualization, Data curation, Formal analysis, Supervision, Funding acquisition, Investigation, Methodology, Writing—original draft, Project administration, Writing—review and editing; DCW, Conceptualization, Formal analysis, Supervision, Funding acquisition, Methodology, Writing—original draft, Project administration

### Author ORCIDs

Rachel Ambler, http://orcid.org/0000-0002-6647-4116
Lea A Hampton-O'Neil, http://orcid.org/0000-0002-9665-170X
Christoph Wülfing, http://orcid.org/0000-0002-6156-9861
David Cameron Wraith, http://orcid.org/0000-0003-2147-5614

### Ethics
Animal experimentation: All animal experiments were carried out under the UK Home Office Project Licence number 30/2705 held by David Wraith and the study was approved by the University of Bristol ethical review committee.

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
