## [Decision Letter]

Thank you for submitting your article "PKCθ links proximal T cell and Notch signaling through localized regulation of the actin cytoskeleton" for consideration by *eLife*. Your article has been favorably evaluated by Tadatsugu Taniguchi (Senior Editor) and three reviewers, one of whom, Michael L Dustin (Reviewer #1), is a member of our Board of Reviewing Editors. The following individual involved in review of your submission has agreed to reveal their identity: Bernard Malissen (Reviewer #2).

The reviewers have discussed the reviews with one another and the Reviewing Editor has drafted this decision to help you prepare a revised submission.

Summary:

Tg4 CD4^+^ T cells isolated from Tg4 mice that are deprived of the serine-threonine kinase PKCq and were injected 18 h previously with the Tg4 cognate antigen showed a reduction in intracellular Notch1 expression as compared to Tg4 CD4^+^ T cells isolated from WT mice. A finding further confirmed by analysis of Notch1-dependent hes1 expression. There is a concern regarding the level of Hes1 activation in relation to cells that utilise bona fide ligand dependent Notch signalling, such as thymocytes, and using thymocytes from the same mouse strains as positive controls would be helpful to put this in perspective. Following CD3 + CD28 stimulation for 18 hours, naive CD4^+^ T cells from B10.PL mice deprived of PKCq showed reduced Notch1 cleavage. This cleavage occurred within a time frame compatible with TCR signalling. T cell proliferation, CD69 and c-Myc upregulation and Th17 development were defective in Tg4KO T cells whereas IL-2 and Th1 cell differentiation were unaffected. Importantly, constitutive activation of Notch in Tg4KO cells restored T cell proliferation and Th17 cell differentiation. The authors admit that this is an over-expression experiment, but it would be helpful to see the level of expression, normalised to cell number, for early DN thymocytes, mature T cells undergoing activation and the retrovirally transduced cells. Visualization of ADAM10 at the T cell-APC interface showed that it was recruited rapidly and transiently to interfaces involving of Tg4WT T cells but not Tg4KO T cells. Therefore, the actin-dependent (Jasplakinolide sensitive) early recruitment of ADAM10 to the T cell-APC interface likely contributes to efficient Notch1 activation downstream of PKC There is a concern as to how jJasplakinolide, a non-specific F-actin stabilizer, can be used in this manner without inhibiting synapse formation – it would be useful to see titration data to better understand the degree to which the width of the optimum Jasplakinolide concentration for this effect θ. Treating Tg4^WT^ T cells with PMA and Calyculin A – a phosphatase inhibitor – resulted in PKCq-dependent Coronin1A phosphorylation. There was a concern that a Calyculin A alone condition was not shown and whether or not PMA is required to see the increase phosphorylation. This gel shown is also sub-optimal due to artefacts and repeating this with the additional condition will also provide an opportunity to improve the quality of the data. This led the authors to propose that PKCθq enhances T cell actin dynamics through co-localization with and phosphorylation of the negative actin regulator Coronin1A, thereby mediating actin-based recruitment of ADAM10 to the T cell-APC interface and efficient Notch activation.

I do not have any specific concerns. The data are solid and the proposed model interesting in that it concurs to integrate the TCR-CD28 signaling pathways with the Notch pathway. As shown by the authors this is of biological relevance since impairment of such integrated loop affects Th17 differentiation. Importantly those studies were performed in primary T cells and not transformed T cells.

Overall, the model is interesting and some of the data are generally convincing, but the series of experiments is marred by use of different time points without explanation. Further explanation of these choices and how this relates to the operation of the proposed mechanism in the line of T-APC interaction would strengthen the work.

Essential revisions:

1) Provide data on Notch ICD and Hes1 levels in mature T cells undergoing activation, DN thymocytes, which are known to use the pathway and show clear Hes1 signatures in public data, and the NICD over-expressing cells used later on. This data could be supplemental, its significance should be clearly discussed.

2) In relation to evidence for CoroninA phosphorylation by PKCtheta, it is a concern that the calyculin A alone condition is omitted. The authors should repeat the experiment with this condition included. How would the authors interpret the result if PMA is not needed to see the effect? The gel shown is also suboptimal and an effort should be made in this new experiment to reduce artefacts that make the interpretation ambiguous.

3) To test whether ADAM10 recruitment to the synapse is mediated by PKC-theta dependent actin changes, they use Jasplakinolide. But this drug would also be expected to impair synapse formation by stabilising F-actin which is then collapsed by myosin II based contractility. Its assumed that this rescue effect is seen only in a narrow range of Jasplakinolide concentrations to semiselectively counteract the loss of CoroninA inhibition. The authors should show a Jasplakinolide titration to clarify this and also it would be helpful to see the total F-actin at each point to understand that nature of this effect. Otherwise it’s hard to understand how a saturating concentration of Jasplakinolide would accomplish the desired effect. The authors could discuss if there is any reason to expect that Jasplakinolide would specifically counteract CoroninA function.

4) The mix of time scales from minutes to hours to days makes it difficult to parse out exactly the kinetics of the mechanism. The authors need to provide a rationale for these timings and how these relate to T-APC interaction time lines that are established in the literature. When is this effect most important?

---

## [Author Response]

*Essential revisions:*

*1) Provide data on Notch ICD and Hes1 levels in mature T cells undergoing activation, DN thymocytes, which are known to use the pathway and show clear Hes1 signatures in public data, and the NICD over-expressing cells used later on. This data could be supplemental, its significance should be clearly discussed.*

Reviewers requested the inclusion of supplementary data on Notch ICD and *Hes1* levels in mature T cells, undergoing activation, in relation to DN thymocytes and cells overexpressing NICD following retroviral transduction. Figure 1—figure supplement 1 and B present data representing 4 experiments in which the influence of T cell activation on Notch ICD (NICD) expression was compared in DN thymocytes, naïve and previously primed T cells from Tg4 mice. NICD levels are constitutively high in DN thymocytes and barely affected by cell activation with anti-CD3 antibodies, presumably because DN thymocytes are either CD3 low or negative. Naïve T cells upregulate NICD on activation while this is less evident in previously primed Tg4 cells. *Hes 1* gene expression is a downstream consequence of the generation of NICD; therefore, we focused on NICD generation as the direct readout for our supplementary experiments. Figure 1—figure supplement 1 and D reveal levels of NICD expressed following retroviral transduction of Tg4 T cells. Clearly, retroviral transduction generates cells with high levels of NICD reproducibly.

*2) In relation to evidence for CoroninA phosphorylation by PKCtheta, it is a concern that the calyculin A alone condition is omitted. The authors should repeat the experiment with this condition included. How would the authors interpret the result if PMA is not needed to see the effect? The gel shown is also suboptimal and an effort should be made in this new experiment to reduce artefacts that make the interpretation ambiguous.*

The Coronin 1A phosphorylation data has been repeated to provide clearer primary data and to include further controls. Data shown in Figure 4 reveals that cell activation with PMA leads to phosphorylation of Coronin 1A; however, this is only observed in samples treated with the phosphatase inhibitor Calyculin A. Treatment of cells with Calyculin A alone was without effect. Notably, Coronin 1A phosphorylation observed in WT Tg4 cells was markedly diminished in CD4^+^ T cells from PKCθ knockout mice (Figure 4).

*3) To test whether ADAM10 recruitment to the synapse is mediated by PKC-theta dependent actin changes, they use Jasplakinolide. But this drug would also be expected to impair synapse formation by stabilising F-actin which is then collapsed by myosin II based contractility. Its assumed that this rescue effect is seen only in a narrow range of Jasplakinolide concentrations to semiselectively counteract the loss of CoroninA inhibition. The authors should show a Jasplakinolide titration to clarify this and also it would be helpful to see the total F-actin at each point to understand that nature of this effect. Otherwise it’s hard to understand how a saturating concentration of Jasplakinolide would accomplish the desired effect. The authors could discuss if there is any reason to expect that Jasplakinolide would specifically counteract CoroninA function.*

Reviewers have requested further data showing titration of the Jasplakinolide drug. As previously communicated with the editorial office, Christoph Wülfing and colleagues recently published a paper in which they conducted a titration of this drug using similar TCR transgenic T cells (5CC7). This paper (Roybal et al. PLOS ONE 10 (8) e0133231) addresses the questions raised by the reviewer. Furthermore, we have shown that 5CC7 and Tg4, the TCR transgenic used in this study, behave identically in cell coupling assays when treated with the drug. The cell coupling-frequencies of Jasplakinolide-treated Tg4 cells, shown relative to 5CC7, are added in the text and are displayed in Figure 3—figure supplement 1. The relevance of the comparative data is referred to in the legend to Figure 3—figure supplement 1. In summary, the cell coupling data shown in this figure establish that Tg4 and 5C.C7 T cells respond comparably to increasing concentrations of Jasplakinolide. The extensive characterization of the effect of 40 nM Jasplakinolide on T cell actin dynamics in (Roybal et al., 2015a) therefore applies to the Tg4^WT^ T cells.

*4) The mix of time scales from minutes to hours to days makes it difficult to parse out exactly the kinetics of the mechanism. The authors need to provide a rationale for these timings and how these relate to T-APC interaction time lines that are established in the literature. When is this effect most important?*

The reviewers have asked us to discuss the mix of time scales from minutes to hours to days. We have included a short paragraph at the end of the paper that emphasises how signalling events of relatively short duration lead to longer-term cellular changes. The extremely rapid signalling events taking place at the immune synapse are best revealed by the type of spatio-temporal analysis used in this paper. Longer-term changes in cell function i.e. cell activation, surface phenotype, cytokine production require medium term analysis, on the scale of hours, while changes in cell differentiation require days to be observed. We have defined the role of PKCθ on such cellular events using techniques that are appropriate for the time scale of the specific event.